# Metabolic capabilities of key rumen microbiota drive methane emissions in cattle

Wanxin Lai,[1] Antton Alberdi,[2] Andy Leu,[3] Arturo V. P. de Leon,[4] Carl M. Kobel,[4] Velma T. E. Aho,[4] Rainer Roehe,[5] Phil B. Pope,[1,3,4] Torgeir R. Hvidsten[1]

**ABSTRACT** The rumen microbiome plays a critical role in determining feed conversion and methane emissions in cattle, with significant implications for both agricultural productivity and environmental sustainability. In this study, we applied a hierarchical joint species distribution model to predict directional associations between biotic factors and abundances of microbial populations determined via metagenome-assembled genomes (MAGs). Our analysis revealed distinct microbial differences, including 191 MAGs significantly more abundant in animals with a higher methane yield (above 24 g/kg dry matter intake [DMI]; high-emission cattle), and 220 MAGs more abundant in low-emission cattle. Interestingly, the microbiome community of the low-methane-emission rumen exhibited higher metabolic capacity but with lower functional redundancy compared to that of high-methane-emission cattle. Our findings also suggest that microbiomes associated with low methane yields are prevalent in specific functionalities such as active fiber hydrolysis and succinate production, which may enhance their contributions to feed conversion in the host animal. This study provides an alternate genome-centric means to investigate the microbial ecology of the rumen and identify microbial and metabolic intervention targets that aim to reduce greenhouse gas emissions in livestock production systems.

**IMPORTANCE** Ruminant livestock are major contributors to global methane emissions, largely through microbial fermentation in the rumen. Understanding how microbial communities vary between high- and low-methane-emitting animals is critical for identifying mitigation strategies. This study leverages a genome-centric approach to link microbial metabolic traits to methane output in cattle. By reconstructing and functionally characterizing hundreds of microbial genomes, we observe that a low-methane-emission rumen harbors well-balanced, "streamlined" microbial communities characterized by high metabolic capacity and minimal metabolic overlap across populations (low functional redundancy). Our results demonstrate the utility of genome-level functional profiling in uncovering microbial community traits tied to climate-relevant phenotypes.

**KEYWORDS** MAGs, Bayesian modeling, joint species distribution models (HMSC), microbiome plasticity, feed conversion, rumen microbiome, methane emissions, microbial diversity, livestock sustainability

Ruminants host a specialized gut microbiome composed of bacteria, fungi, viruses, and archaea, which ferments fibrous feed into energy-yielding short-chain fatty acids, but also methane ($CH_4$), a major greenhouse gas (1). Rumen microorganisms can be broadly categorized as generalists that utilize a wide array of substrates, or specialists, which occupy specific metabolic niches, such as methanogens (2). The balance between generalists and specialists is believed to influence both microbiome

Address correspondence to Wanxin Lai, wanxin.lai@nmbu.no.

The authors declare no conflict of interest.

See the funding table on p. 6.

plasticity (e.g., flexibility) and metabolic efficiency in the rumen microbiome, ultimately affecting phenotypes such as methane emissions. Archaeal and bacterial microbiomes with higher diversity and functional redundancy tend to be more stable but have less plasticity, whereas lower-diversity systems tend to be more adaptable to environmental shifts (3, 4). Specifically for dairy cattle, lower microbiome richness, estimated via gene content and 16S-rRNA-based structural analysis, has been tightly linked to higher feed efficiency and lower methane potential (5). Herein, we sought to explore how microbiome plasticity, assessed via functionally characterized microbial genomes, changes in cattle with varying methane emissions (measured as g/kg dry matter intake [DMI]).

To explore microbial community responses, we fit a Hierarchical Modeling of Species Communities (HMSC) model to the data (6), associating metagenome-assembled genome (MAG) abundances with experimental variables. This analysis was based on 700 high-quality MAGs reconstructed from 27 rumen samples collected at five timepoints from three high-emission cattle (HEC) and three low-emission cattle (LEC), classified based on a 24 $CH_4$ (g/kg DMI) threshold. Our genome-centric approach leverages strain-level functional characterization, allowing us to quantify metabolic capacity indices (MCIs) derived from pathway annotations, rather than relying solely on taxonomic and gene-based summaries. The model achieved good convergence (potential scale reduction factor ~1) and effective sample sizes (ESS > 200) across parameters, supporting robust inference on the ecological drivers of MAG abundance. Phylogenetic clustering incorporated in the model revealed two clades: Clade 1, dominated by *Bacteroidota,* was more abundant in LEC, whereas Clade 2, encompassing various phyla, was more abundant in HEC (CH4Low; Fig. 1B). A strong phylogenetic signal, inherent to the model, linked methane yield to MAG abundances, with Clade 1 MAGs predominant in LEC (Rho = 1 [0.99, 1]) and Clade 2 MAGs prevalent in HEC (Rho = 0.99 [0.99, 1]). This identified 191 MAGs significantly associated (≥90% posterior probability) with HEC (i.e., with higher abundance in HEC) and 220 MAGs with LEC (Fig. 1C). Of the variance in MAG abundance explained by the model (26.2%), 13.6% was explained by methane yield (Table S1).

## METABOLIC EFFICIENCY AND MICROBIAL COMPOSITION VARY ACROSS HIGH AND LOW EMITTING CATTLE

The rumen microbiome of LEC was enriched in populations affiliated to the *Bacteroidota* and *Actinobacteria* (Fig. 1C) that collectively encode lower functional redundancy (Fig. 2C). To explore this further, we associated genome-inferred functional traits with methane yield using HMSC (Fig. 2B). The rumen microbiome of LEC was predicted to encode a higher metabolic capacity for degradation of starch and plant fiber commonly ingested by cattle fed a mixed forage and concentrate diet, including cellulose, xyloglucans, alpha-galactans, and xylans. Moreover, higher abundances of proficient rumen fibrolytic microbes (e.g., *Fibrobacter succinogenes*, MCI 0.27) were also observed (Table S3), along with specific synergistic partners such as *Prevotella ruminicola* (MCI 0.27), which have been previously identified via co-culture to enhance fiber hydrolysis and succinate production (9). Concurrently, we predicted a higher capacity of LEC-MAGs to produce succinate (Fig. 2B). We also observed fewer key enzyme domains for lactate metabolism and acrylate pathway in LEC than in HEC (EC 2.8.3.1, EC 1.3.8.7, EC 4.2.1.54) (Fig. S3 and S4), along with less detectable lactate accumulation (5), suggesting that the LEC community in this study is less reliant on lactate metabolism and acrylate-based propionate production.

The rumen microbiome of HEC was characterized by a higher *Bacillota*-to-*Bacteroidota* ratio and an increased abundance of *Methanobacteriota*, while the microbiome community displayed higher functional redundancy (Fig. 2C), which, based on prior knowledge, would suggest a lower metabolic efficiency (12). In general, higher functional redundancy is suggestive of greater metabolic stability by allowing multiple microbial species to perform overlapping roles with less metabolic versatility (Fig.

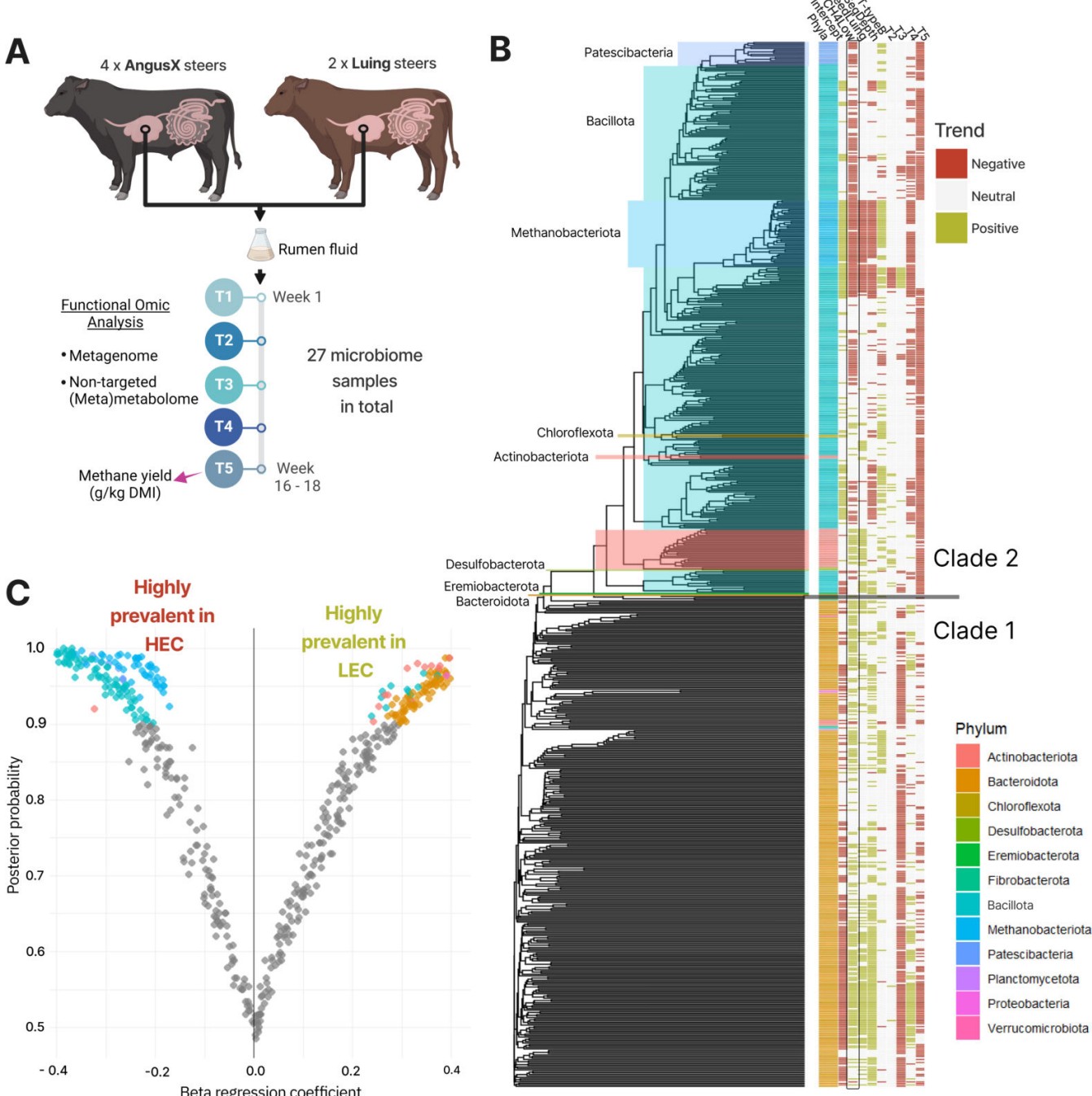

**FIG 1** (A) Experimental design: Our approach leveraged 700 MAGs reconstructed from rumen fluid samples collected across five time points from six cattle (two breeds: Aberdeen Angus X and Luing; 27 samples total; Table S5) fed with a mixed forage and concentrate diet. Animals exhibiting variable methane emission levels (24 CH₄ [g/kg DMI] as the cutoff point for high and low emissions). The genomes were reconstructed with both long- and short-read sequencing, functionally annotated via DRAM (7) and "distilled" into genome-inferred functional traits —MCI, by means of KEGG and MetaCyc metabolic pathway fullness values using distillR. All methods concerning the animal trial, rumen sampling, as well as metagenomic and metabolomic data generation are presented (8). Code for data analysis can be found at https://gitlab.com/wanxin.lai/metaG-SuPacow.git. (B) Phylogenetic tree and heatmap displaying the responses of MAGs to experimental variables modeled by HMSC, incorporating fixed effect in column order: methane emission (CH4Low), breed, sequencing depth (log-scale), community type (RCT-type) (8) and sampling time (T2, T3, T4, T5), details in Fig. S2. Host identity (individual cow ID) was included as a random effect. Positive (green) and negative (red) trends indicate where MAGs are more abundant, for example, the negative trend of Methanobacteriota with CH4Low means that these are more abundant in HEC than LEC. Although "Time" is not the focus, MAG responses across timepoints indicate which phyla thrive in cattle

Fig 1 (Continued)

rumen over the long term. Most phyla exhibit a declining trend by T5, except Methanobacteriota (Clade 2), showing a neutral trend at T5 compared to T1, indicating stable abundance over time. Strain-level metabolic characterization of the two clades captured a complex functional landscape, in which some taxa (Methanobacteriota, Patescibacteria) formed a distinct functional cluster while others (Bacillota, Actinobacteriota, Bacteroidota) depicted a wider spectrum of metabolic traits (Fig. S1). (C) Volcano plot showing regression coefficients of MAG abundances associated with methane emission (CH4Low: LEC = 1, HEC = 0). MAGs with ≥90% posterior probability supporting an association with emission levels are colored by phylum.

2C). Given that accumulated evidence has linked low methane-emitting animals with increased lactate metabolism (13), we were surprised to observe increased capacity for HEC-MAGs to produce lactate, as well as increased copies of L-lactate dehydrogenase

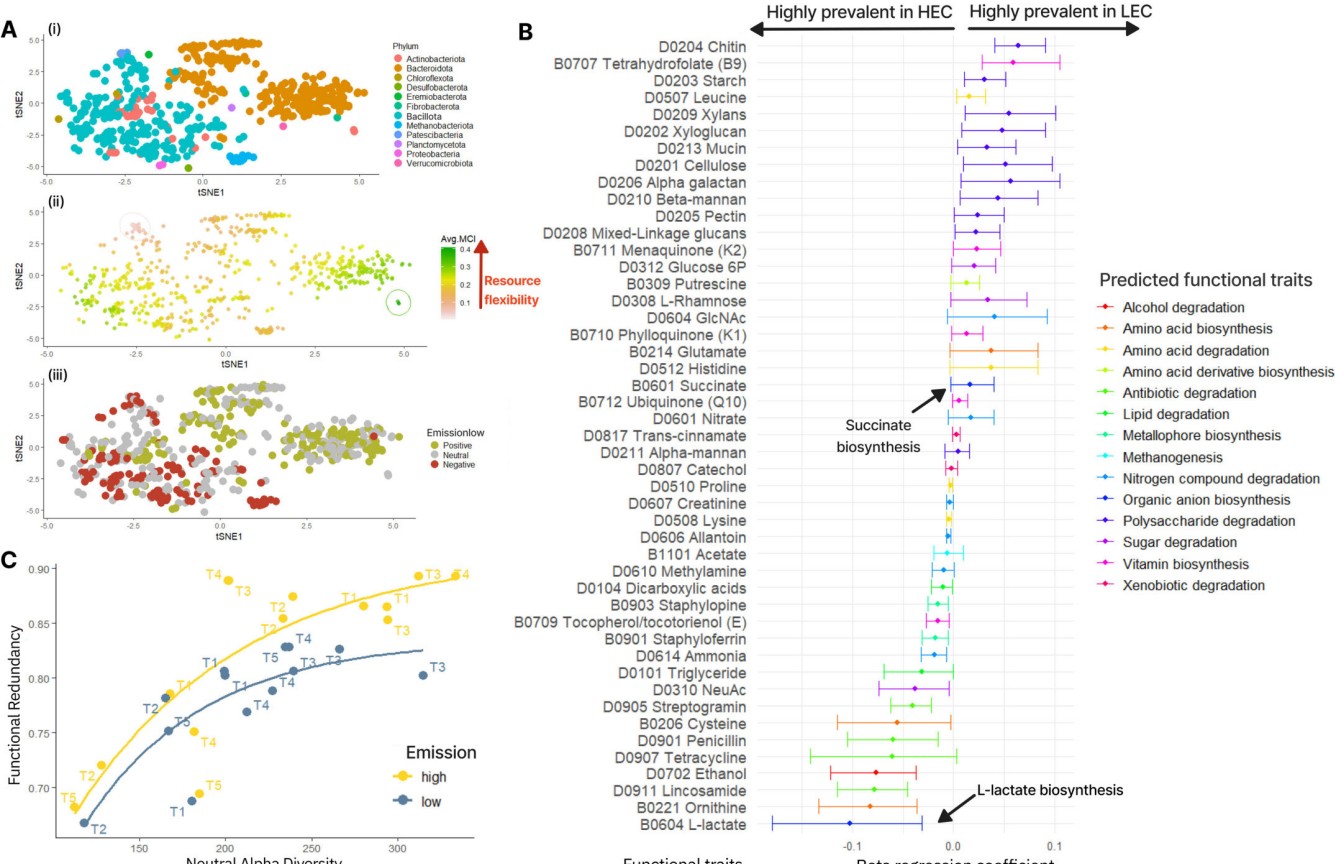

FIG 2 (A) t-SNE plots showing ordination of MAG abundance data, colored by (i) bacterial phyla, (ii) average metabolic capacity (MCI), and (iii) HMSC-modeled associations of MAGs with LEC (green), HEC (red), or neutral emissions (gray). MCI represents the relative proportion of biosynthetic and degradative genes in each genome. In (ii), lower MCI values (~0, pink) indicate niche-specialized microbes. For example, MAGs colored red within the pink circle, from the Patescibacteria (genus UBA2834, Nanosyncoccus; MCI approximately 0.03), which have the lowest MCI, are prevalent in HEC. In contrast, higher values (~0.4, dark green) suggest metabolic versatility. *Thermobifida fusca* and *Nocardiopsis alba* (Actinobacteriota) exhibited the highest MCI (>0.4) but showed no emission-specific abundance trend (green circle). (B) Top predicted functional traits (BH-adjusted *P* < 0.05) differentiating rumen microbial communities. Error bars show 90% quantile intervals across 600 posterior estimations, with non-overlapping intervals highlighting key functional differences. We predicted a higher capacity of LEC-MAGs to produce succinate, which was supported by more observed copies of putative enzyme domains (EC 6.4.1.3, EC 5.1.99.1, EC 2.8.3.27, EC 4.1.1.-, EC 5.4.99.2) for the methylmalonyl–CoA pathway (part of the succinyl-CoA production in Fig. S3 and S4). This suggests that the LEC microbial community favors polysaccharide degradation and alternative hydrogen sinks, such as succinate (a precursor to propionate), which is supported by the higher observed propionate accumulation and may have implications for methane reduction in LEC (Fig. S3) (10, 11). (C) Relationship between neutral alpha diversity and functional redundancy, showing a positive correlation, HEC microbiomes exhibit a higher curve than LEC, reflecting greater functional redundancy at higher alpha diversity levels. In contrast, LEC microbiomes, despite lower diversity, maintain function through metabolic versatility, suggesting adaptability to resource variability. Neutral alpha diversity captures species richness without weighting by phylogenetic distance or functional traits; it serves as a baseline for community diversity, which we can evaluate whether increased species richness correlates with functional redundancy.

(EC 1.1.1.27) (Fig. S3 and S5). In support of this, we also noted increased MAG abundances in HEC affiliated to several renowned lactate producers: *Streptococcus* spp. (MCI 0.14), *Bifidobacterium ruminantium, Kandleria vitulina,* and *Sharpea azabuensis* (MCI ≥ 0.22), sharing the same metabolic role (Table S3) (10). However, we did not detect key metabolisms nor populations that are reputed to convert lactate to butyrate and/or propionate (e.g., *Megasphaera* and *Coprococcus* spp.). An absence of lactate utilizers was also in line with higher observed lactate accumulation in metabolomic analysis (Fig. S3) and suggestive of fermentative limitations in HEC from this study.

Collectively, our results highlight that a low-methane-emission rumen harbors microbial communities characterized by high metabolic capacity (20.3% higher MCI compared to HEC, *P*-value = 2.056e-09, Table S2) and minimal metabolic overlap across populations (low functional redundancy). In this context, we speculate that a well-balanced, high-capacity yet "streamlined" microbiome is reflective of core metabolic pathways that are highly adaptable to resource variability found in the mixed forage and concentrate diet used in this study. In LEC, distinct metabolic strategies were evident in the utilization of complex plant polysaccharides and succinate-to-propionate metabolism, frequently observed in low-methane-emitting microbiomes (14). Conversely, HEC in this study exhibited higher acrylate-CoA and lactate levels. The scarcity of HEC microbiota predicted to perform lactate-driven hydrogen sequestration to propionate and/or butyrate is suggestive of fermentation inefficiencies that would reduce competition for hydrogen and potentially increase its availability for methanogens.

We were highly encouraged that our genome-centric analyses performed herein have largely mirrored previous gene- and taxonomy-based studies that have linked reduced gene and taxonomic richness to high feed efficiency and low methane production. Moreover, deeper genome inference of functional traits reiterated findings from multiple studies that have shown ruminal fermentation via succinate-to-propionate is prominent in LEC microbiomes. We acknowledge the limited sample size used in this study restricts broader biological interpretations; however, this approach is highly amendable to scale and will be increasingly applicable as the rumen microbiome field continues its shift toward a genome-centric methodology.

## ACKNOWLEDGMENTS

We gratefully acknowledge the financial support of the Novo Nordisk Foundation under 0054575-SuPAcow. This project has received funding from the European Union's Horizon 2020 Research and Innovation programme under grant agreement number No.101000213. P.B.P. also acknowledges support from the Australian Research Council (Future Fellowship: FT230100560). The authors acknowledge the Orion High-Performance Computing Center at the Norwegian University of Life Sciences and Sigma2—the National Infrastructure for High-Performance Computing and Data Storage in Norway for providing computational resources that have contributed to computations reported in this paper. We also acknowledge Elixir Norway, supported by the Research Council of Norway's (NFR) grant 322392, for the bioinformatics and data management support received for this paper. The authors further acknowledge financial support from the Scottish Government (RESAS Division) and Biotechnology and Biological Sciences Research Council (BBSRC BB/S006567). We also thank the staff of the SRUC Beef Research Center for their excellent technical support.

## AUTHOR AFFILIATIONS

[1]Faculty of Chemistry, Biotechnology and Food Science, Norwegian University of Life Sciences, Ås, Norway
[2]Center for Evolutionary Hologenomics, Globe Institute, University of Copenhagen, Copenhagen, Denmark

³Centre for Microbiome Research, Faculty of Health, School of Biomedical Sciences, Queensland University of Technology, Translational Research Institute, Woolloongabba, Australia
⁴Faculty of Biosciences, Norwegian University of Life Sciences, Ås, Norway
⁵Department of Agriculture, Horticulture and Engineering Sciences, Scotland's Rural College, Edinburgh, United Kingdom

**AUTHOR ORCIDs**

Wanxin Lai  http://orcid.org/0009-0000-1607-2453
Antton Alberdi  http://orcid.org/0000-0002-2875-6446

**FUNDING**

| Funder | Grant(s) | Author(s) |
| --- | --- | --- |
| Novo Nordisk Fonden | 0054575-SuPAcow | Phil B. Pope |
| HORIZON EUROPE European Research Council | 101000213 | Phil B. Pope |
| Australian Research Council | FT230100560 | Phil B. Pope |

**AUTHOR CONTRIBUTIONS**

Wanxin Lai, Conceptualization, Data curation, Formal analysis, Methodology, Software, Visualization, Writing – original draft, Writing – review and editing | Antton Alberdi, Methodology, Validation, Visualization, Writing – review and editing | Andy Leu, Resources | Arturo V. P. de Leon, Resources | Carl M. Kobel, Resources | Velma T. E. Aho, Conceptualization, Resources | Rainer Roehe, Conceptualization, Resources, Validation | Phil B. Pope, Conceptualization, Funding acquisition, Project administration, Resources, Supervision, Validation, Writing – review and editing | Torgeir R. Hvidsten, Methodology, Resources, Supervision, Validation, Writing – review and editing

**DATA AVAILABILITY**

The data generated in this study have been deposited in the European Nucleotide Archive (ENA) database under accession code PRJEB83989. Animal metadata and all processed omics data, including metagenome-assembled genomes (MAGs), are available through the Norwegian Research Infrastructure Services at https://ns9864k.web.sigma2.no/TheMEMOLab/projects/SupaCow/data_for_publication/metagenomics/.

**ETHICS APPROVAL**

The animal experiment was conducted at the Beef and Sheep Research Center of Scotland's Rural College (six miles south of Edinburgh, UK). The experiment was approved by the Animal Experiment Committee of SRUC and was conducted in accordance with the requirements of the UK Animals (Scientific Procedures) Act 1986.

**ADDITIONAL FILES**

The following material is available online.

Supplemental Material

**Supplemental material (mSystems00601-25-s0001.pdf).** Supplemental figures and tables.

Open Peer Review

**PEER REVIEW HISTORY (review-history.pdf).** An accounting of the reviewer comments and feedback.

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
