## [Reviewer comments · mSystems]

Metabolic Capabilities of Key Rumen Microbiota Drive Methane Emissions in Cattle

Wanxin Lai, Antton Alberdi, Arturo de León, Carl Kobel, Velma Aho, Andy Leu, Rainer Roehe, Phillip Pope, and Torgeir Hvidsten

Corresponding Author(s): Wanxin Lai, Norges miljø- og biovitenskapelige universitet Fakultet for kjemi bioteknologi og matvitenskap

Review Timeline:

Submission Date:	April 28, 2025
Editorial Decision:	August 22, 2025
Revision Received:	August 26, 2025
Accepted:	August 28, 2025

Editor: Marnix Medema

Reviewer(s): The reviewers have opted to remain anonymous.

Transaction Report:

DOI: <https://doi.org/10.1128/msystems.00601-25>

Re: mSystems00601-25 (Metabolic Capabilities of Key Rumen Microbiota Drive Methane Emissions in Cattle)

Dear Miss Wanxin Lai:

As you can see, the reviewers only have minor feedback. Please address carefully, and I expect to be able to move forward with your paper without further re-review.

Please return the manuscript within 30 days; if you cannot complete the modification within this time period, please contact me. If you do not wish to modify the manuscript and prefer to submit it to another journal, notify me immediately so that the manuscript may be formally withdrawn from consideration by mSystems.

Revision Guidelines

Sincerely,
Marnix Medema
Editor
mSystems

Reviewer #1 (Comments for the Author):

The authors present an interesting observation that specific MAG's in cattle rumen are related to methane emissions. They also found that MAG's related to low methane emissions exhibited higher metabolic capacity, but lower redundancy. Overall I think this paper is an interesting observation with high public interest because reducing cattle methane emissions is important for reducing the carbon footprint of cattle. Some minor comments.

1. I did not see any information about IACUC approval. I think this needs to be added or at least explicitly stated to be in the publication the data is from.
2. I would like more details as to the distribution of each breed that were low versus high emission. Although only 6% of the variance in the model is explained by the breed I would at least like to know if the high versus low emission groups are evenly distributed across the breeds.
3. Line 63: I think it might be fit instead of fitted.

Reviewer #2 (Comments for the Author):

The objective of this text was to explore how microbiome plasticity assessed via functionally characterized microbial genomes changes in cattle with varying methane emission using a hierarchical joint species distribution model. This model can predict directional associations and not only correlation between biotic factors and abundance of microbial population. The aim of the text is very original. It gave relevant conclusions in line with previous knowledge on the link between reduced genes and taxonomic richness to low methane production and will provide a new mean to investigate microbial ecology of the rumen leading to potential broader biological interpretations. The text is very well written with clear analyses and interpretation of the data.

My only comment is that the choice of neutral diversity to evaluate the relationship between diversity and functional redundancy need to be explained.

MSystems00601-25

Title: Metabolic Capabilities of Key Rumen Microbiota Drive Methane Emissions in Cattle

By: Wanxin Lai, Antton Alberdi, Andy Leu, Arturo V. P. de Leon, Carl M. Kobel, Velma T. E. Aho, Rainer Roehe, Phil B. Pope, Torgeir R. Hvidsten

General comments:

The objective of this text was to explore how microbiome plasticity assessed via functionally characterized microbial genomes changes in cattle with varying methane emission using a hierarchical joint species distribution model. This model can predict directional associations and not only correlation between biotic factors and abundance of microbial population. The aim of the text is very original. It gave relevant conclusions in line with previous knowledge on the link between reduced genes and taxonomic richness to low methane production and will provide a new mean to investigate microbial ecology of the rumen leading to potential broader biological interpretations. The text is very well written with clear analyses and interpretation of the data.

My only comment is that the choice of neutral diversity to evaluate the relationship between diversity and functional redundancy need to be explained.

L28 why do you use the word “also”? I recommend to delete it.

Reviewer #1 (Comments for the Author):

1. I did not see any information about IACUC approval. I think this needs to be added or at least explicitly stated to be in the publication the data is from.

We thank the reviewer for carefully noting this point. To clarify this for readers, we have now added the following statement to the Ethics Declarations section of the manuscript (see line 214th in the marked-up file) :

The animal experiment was conducted at the Beef and Sheep Research Center of Scotland's Rural College (6 miles south of Edinburgh, UK). The experiment was approved by the Animal Experiment Committee of SRUC and was conducted in accordance with the requirements of the UK Animals (Scientific Procedures) Act 1986.

2. I would like more details as to the distribution of each breed that were low versus high emission. Although only 6% of the variance in the model is explained by the breed I would at least like to know if the high versus low emission groups are evenly distributed across the breeds.

A table addressed this information has been added into supplemental file as Table S5 (see line 87th in the marked-up file).

3. Line 63: I think it might be fit instead of fitted.

Thank you, it has been corrected.

Reviewer #2 (Comments for the Author):

My only comment is that the choice of neutral diversity to evaluate the relationship between diversity and functional redundancy need to be explained.

Thanks for the review and comment. We have clarified that neutral alpha diversity provides a baseline expectation which enables us to assess the relationship between observed diversity and functional redundancy beyond what would be expected by chance. See line 163rd in the marked-up file.

Re: mSystems00601-25R1 (Metabolic Capabilities of Key Rumen Microbiota Drive Methane Emissions in Cattle)

Dear Miss Wanxin Lai:

Your manuscript has been accepted, and I am forwarding it to the ASM production staff for publication. Your paper will first be checked to make sure all elements meet the technical requirements. ASM staff will contact you if anything needs to be revised before copyediting and production can begin. Otherwise, you will be notified when your proofs are ready to be viewed.

Data Availability: ASM policy requires that data be available to the public upon online posting of the article, so please make sure all sequence data is uploaded to public repositories with corresponding accession numbers, verify all links to sequence records, if present, and make sure that each number retrieves the full record of the data. If a new accession number is not linked or a link is broken, provide production staff with the correct URL for the record. If the accession numbers for new data are not publicly accessible before the expected online posting of the article, publication may be delayed; please contact ASM production staff immediately with the expected release date.

Sincerely,
Marnix Medema
Editor
mSystems